# Navigating the Landscape of Cardiovascular Risk Scores: A Comparative Analysis of Eight Risk Prediction Models in a High-Risk Cohort in Lithuania

**DOI:** 10.3390/jcm13061806

**Published:** 2024-03-21

**Authors:** Petras Navickas, Laura Lukavičiūtė, Sigita Glaveckaitė, Arvydas Baranauskas, Agnė Šatrauskienė, Jolita Badarienė, Aleksandras Laucevičius

**Affiliations:** 1Faculty of Medicine, Institute of Clinical Medicine, Vilnius University, 03101 Vilnius, Lithuania; lukaviciute.laura@gmail.com (L.L.); sigita.glaveckaite@santa.lt (S.G.); arvydas.baranauskas@santa.lt (A.B.); agne.satrauskiene@gmail.com (A.Š.); jolita.badariene@santa.lt (J.B.); 2State Research Institute Centre for Innovative Medicine, 08410 Vilnius, Lithuania; aleksandras.laucevicius@santa.lt

**Keywords:** cardiovascular risk, risk prediction models, inter-model agreement, Cohen’s Kappa, Lithuanian cohort, clinical decision-making, risk stratification

## Abstract

**Background:** Numerous cardiovascular risk prediction models (RPM) have been developed, however, agreement studies between these models are scarce. We aimed to assess the inter-model agreement between eight RPMs: assessing cardiovascular risk using SIGN, the Australian CVD risk score (AusCVDRisk), the Framingham Risk Score for Hard Coronary Heart Disease, the Multi-Ethnic Study of Atherosclerosis risk score, the Pooled Cohort Equation (PCE), the QRISK3 cardiovascular risk calculator, the Reynolds Risk Score, and Systematic Coronary Risk Evaluation-2 (SCORE2). **Methods:** A cross-sectional study was conducted on 11,174 40–65-year-old individuals with diagnosed metabolic syndrome from a single tertiary university hospital in Lithuania. Cardiovascular risk was calculated using the eight RPMs, and the results were categorized into high, intermediate, and low-risk groups. Inter-model agreement was quantified using Cohen’s Kappa coefficients. **Results:** The study revealed significant heterogeneity in risk categorizations with only 1.49% of cases where all models agree on the risk category. SCORE2 predominantly categorized participants as high-risk (67.39%), while the PCE identified the majority as low-risk (62.03%). Cohen’s Kappa coefficients ranged from −0.09 to 0.64, indicating varying degrees of inter-model agreement. **Conclusions:** The choice of RPM can substantially influence clinical decision-making and patient management. The PCE and AusCVDRisk models exhibited the highest degree of agreement while the SCORE2 model consistently exhibited low agreement with other models.

## 1. Introduction

Cardiovascular diseases (CVD) remain a leading cause of morbidity and mortality worldwide, accounting for an estimated 17.9 million deaths annually, which constitutes approximately 31% of all global deaths necessitating precise and reliable risk assessment tools for effective prevention and management [1]. Cardiovascular risk estimation helps the clinician to adjust the intensity of the preventive efforts and identifies high-risk patients deserving immediate and more aggressive intervention, thus helping to personalize lifestyle and risk factor management [2]. The current guidelines advocate adjusting the intensity of blood pressure- and lipid-lowering treatment based on the patient’s cardiovascular risk profile [3,4]. However, since the risk of future CVD events typically relates to a combination of several risk factors, a clinical estimation of these combined effects is somewhat unreliable [5]. As a result, numerous cardiovascular risk prediction models (RPM) have been developed to stratify individuals into various risk categories [6]. Although a substantial literature exists regarding the prognostic values of these scores, studies focusing exclusively on the agreement between risk categories are comparatively scarce and thus represent a domain warranting further scientific inquiry. The present study aims to fill this gap by conducting a comprehensive comparison of eight widely-used cardiovascular risk prediction models: assessing cardiovascular risk using SIGN (ASSIGN), the Australian CVD risk score (AusCVDRisk), the Framingham Risk Score for Hard Coronary Heart Disease (FRS-hCHD), the Multi-Ethnic Study of Atherosclerosis risk score (MESA), the Pooled Cohort Equation (PCE), the QRISK3 cardiovascular risk calculator (QRISK3), the Reynolds Risk Score (RRS), and Systematic Coronary Risk Evaluation 2 (SCORE2).

## 2. Materials and Methods

### 2.1. Study Population/Inclusion and Exclusion Criteria

The dataset comprised patients from the Lithuanian High Cardiovascular Risk (LitHiR) primary prevention program, which is a state-funded program in Lithuania that was started in 2006 with a strong multifactorial focus on middle-aged high cardiovascular (CV) risk subjects to prevent early atherosclerosis development [7]. The primary prevention program included individuals between 40 and 65 years of age without overt cardiovascular disease, diagnosed with metabolic syndrome (MetS), who were investigated between 2006 and 2023 in a large tertiary hospital—Vilnius University Hospital Santaros Klinikos, Vilnius, Lithuania. The MetS was defined according to the revised National Cholesterol Education Program Adult Treatment Panel III (NCEP ATPIII) criteria, meeting three or more criteria: waist circumference ≥ 102 cm in men and ≥88 cm in women; systolic blood pressure (SBP) ≥ 130 mmHg and/or diastolic blood pressure (DBP) ≥ 85 mmHg, or if the diagnosis of hypertension was documented in the medical record; fasting plasma glucose ≥ 5.6 mmol/L or type 2 diabetes mellitus; triglyceride (TG) concentration ≥ 1.7 mmol/L or special treatment is administered to reduce TG concentration; and high-density lipoprotein (HDL) cholesterol < 1.03 mmol/L in men and <1.29 mmol/L in women [8].

The data collected in a prospective and standardized fashion within the principal LitHiR program served as the source from which the essential variables for the current study were extracted. Subjects were considered eligible for inclusion if they had recorded values for low-density lipoprotein (LDL), HDL, total cholesterol (TC), SBP, DBP, fasting glucose, creatinine, C-reactive protein (CRP), and their family history as well as medication history. Subjects missing any of these key parameters were excluded from the study to ensure the comprehensiveness and reliability of the risk stratification. 

The exclusion criteria for this study were previously diagnosed coronary artery disease, silent myocardial ischemia, a transient ischemic attack, an ischemic and hemorrhagic stroke, peripheral artery disease, oncological disorders, advanced kidney or hepatic failure, chronic or persistent arrhythmias, gout, severe psychiatric disorders, as well as pregnancy, drug addiction, and treatment with xanthine oxidase inhibitors.

### 2.2. Risk Prediction Models

#### 2.2.1. Systematic Coronary Risk Evaluation 2 (SCORE2)

SCORE2 is a risk score used to estimate the risk of developing fatal and non-fatal CVD in individuals without previous CVD aged 40–69 years in Europe over a 10-year period. It is an updated version of the original SCORE model, incorporating more contemporary data [9]. It takes into account several risk factors, including gender, age, smoking status, systolic blood pressure, and non-HDL cholesterol. The risk score calculation was based on the following calculator: https://u-prevent.com/calculators/score2 (accessed on 2 June 2023). We utilized a version of the risk calculator that has been recalibrated specifically for regions classified as ‘very high risk’ according to the relevant guidelines. No modifications to the calculation procedure were made.

#### 2.2.2. Pooled Cohort Equations (PCE) Cardiovascular Risk Calculator

PCE estimates an individual’s 10-year risk of developing atherosclerotic cardiovascular disease (ASCVD) based on data from multiple community-based populations, and is applicable to African-American and non-Hispanic White men and women from 40 to 79 years of age [10]. The risk score includes classical cardiovascular risk factors such as age, sex, cholesterol, HDL-cholesterol, systolic blood pressure, hypertension treatment, race, smoking status, and diabetes status. The risk score calculation was based on the following calculator: https://static.heart.org/riskcalc/app/index.html#!/baseline-risk (accessed on 1 June 2023). No modifications to the calculation procedure were made. 

#### 2.2.3. QRISK3 Risk Calculator (QRISK3)

The QRISK3 score was developed in 2017 as an updated QRISK2 algorithm which was published in 2008 and was the standard of care risk tool for the prediction of 10-year risk for cardiovascular events in England’s population aged between 25 and 84 years [11]. The model includes 8 additional risk variables which were identified as possible risk factors of CVD in other studies such as the following: migraine, corticosteroid use, systemic lupus erythematosus, treatment with atypical antipsychotic medications, severe mental illness, erectile dysfunction, and variability of blood pressure. The risk score calculation was based on the following calculator: https://qrisk.org/ (accessed on 2 June 2023). No modifications to the calculation procedure were made.

#### 2.2.4. Framingham Risk Score for Hard Coronary Heart Disease (FRS-hCHD)

The model is designed to estimate the 10-year risk of developing coronary heart disease (CHD) events, such as myocardial infarction and coronary death. It is derived from the Framingham Heart Study and incorporates multiple cardiovascular risk factors including age, sex, TC, HDL cholesterol, SBP, treatment for hypertension, and smoking status. FRS-hCHD is intended for use in non-diabetic patients aged 30–79 years with no prior history of coronary heart disease or intermittent claudication [8]. The risk score calculation was based on the following calculator: https://www.mdcalc.com/calc/38/framingham-risk-score-hard-coronary-heart-disease (accessed on 1 June 2023). No modifications to the calculation procedure were made.

#### 2.2.5. Reynolds Risk Score (RRS)

The RRS is designed to predict the 10-year risk of experiencing a major cardiovascular event (myocardial infarction, stroke, or other major heart disease). The model incorporates traditional risk factors, such as age, sex, blood pressure, cholesterol levels, and smoking status, as well as additional biomarkers such as high-sensitivity CRP and a family history of premature atherosclerosis [12]. Initially developed for women, it was later adapted for men. The risk score calculation was based on the following calculator: http://www.reynoldsriskscore.org/ (accessed on 2 June 2023). No modifications to the calculation procedure were made.

#### 2.2.6. Assessing Cardiovascular Risk Using SIGN (ASSIGN)

The ASSIGN score is designed to estimate the 10-year risk of cardiovascular events in subjects without established CVD by adding social deprivation (Scottish Index of Multiple Deprivation) and family history to the traditional risk factors [13]. It has been shown to be more accurate than other CVD risk scores in predicting CVD risk in the Scottish population. It is also the CVD risk score recommended by the Scottish Intercollegiate Guidelines Network (SIGN) and the Scottish Government Health Directorates. The risk score calculation was based on the following calculator: https://www.assign-score.com/estimate-the-risk/visitors/ (accessed on 2 June 2023). During the calculation procedure, the inclusion of the Scottish index of social deprivation was not applicable. 

#### 2.2.7. Australian CVD Risk Score (AusCVDRisk)

The AusCVDRisk calculator is designed to estimate the 5-year risk of cardiovascular events and is validated for use in people without known CVD aged 30 to 79 years who do not already meet high-risk criteria. The calculator is based on the NZ PREDICT-1° equation, which was developed from a large, contemporary New Zealand primary care cohort study [14]. The equation has been recalibrated to the Australian population and modified for the Australian healthcare system. The risk score calculation was based on the following calculator: https://www.cvdcheck.org.au/calculator (accessed on 2 August 2023). During the calculation procedure, the inclusion of a postcode variable was not applicable. 

#### 2.2.8. Multi-Ethnic Study of Atherosclerosis (MESA) Risk Score

The MESA is designed to estimate the 10-year risk of cardiovascular events in people without known CVD aged 45 to 85 years [15]. It was developed using data from the MESA study, which is a large, multi-ethnic study of over 6800 people. The study has been shown to be accurate in predicting CVD risk in people of all races and ethnicities. The risk score calculation was based on the following calculator: https://www.mesa-nhlbi.org/MESACHDRisk/MesaRiskScore/RiskScore.aspx (accessed on 2 June 2023). The coronary artery calcification index was not included during the calculation procedure since our dataset did not include this variable. 

### 2.3. Variable Definitions

Each RPM possesses unique risk categories and score intervals, necessitating methodological standardization for analytical coherence. Figure 1 visually represents these risk categories and their respective intervals across the different models. For some RPMs, the categorization intervals remained unaltered, while only the nomenclature was adapted. For instance, in the ASSIGN risk score, the original “non-high risk” category was subdivided into low- and intermediate-risk categories for analytical congruence. In other models, such as the PCE, the “borderline risk” label was reclassified as “intermediate risk” to align with the standardized naming convention employed in this study. By standardizing the risk categories, we facilitated a more nuanced comparative analysis.

### 2.4. Statistical Analysis

In the statistical analysis, the level of agreement among the eight cardiovascular risk prediction models was rigorously assessed through a multi-faceted approach. An initial exploration was conducted using descriptive statistics to enumerate the frequency distribution of the risk categories for each model, serving as a foundational layer for understanding how each model stratifies the patient population into distinct risk tiers. To quantitatively measure the concordance between each pair of models, Cohen’s Kappa statistics were employed. This metric was selected for its robustness in assessing agreement in categorical data, providing a range from −1 to 1 where higher values signify stronger agreement.

To enhance the interpretability of the pairwise Kappa values, a heatmap was generated. A further elucidation of the relationships among the models was achieved through hierarchical clustering. Based on the pairwise Kappa statistics, a dendrogram was created using the Ward method, which is effective in minimizing the variance within clusters. This technique aids in grouping models that exhibit analogous risk stratification patterns.

For a more comprehensive view of the inherent structure and variance in the risk categories assigned by these models, Principal Component Analysis (PCA) was applied. This dimensionality reduction technique is adept at summarizing high-dimensional data while preserving their original variance. Lastly, an aggregate view of model agreement was obtained through a Collective Model Agreement Analysis. This involved computing the frequency and percentage of patients for whom a given number of models converged on the same risk category, providing a holistic insight into the level of model concordance in a clinical context.

Statistical analyses were performed using the IBM SPSS software 25.0 version (SPSS, Chicago, IL, USA) or Python, leveraging libraries such as Pandas for data manipulation, scikit-learn for statistical modeling, and Matplotlib and Seaborn for data visualization. When interpreting the results of statistical tests, the level of significance was set to be equal to 0.05.

### 2.5. Ethical Considerations

The research was authorized by the Vilnius Regional Biomedical Research Ethics Committee (permission No. 2019/3-1104-603).

## 3. Results

### 3.1. Descriptive Statistics

In the current study, a total of 11,174 participants were evaluated within the LitHiR cohort. The subjects were predominantly female (*n* = 6527, 58.4%), with a mean age of 53.49 ± 6.47 years. Key cardiovascular risk factors were notably present; the mean body mass index (BMI) was 31.57 ± 4.46 kg/m^2^, and the mean TC level was 6.17 ± 1.37 mmol/L. The lipid profile further indicated a mean LDL cholesterol of 3.98 ± 1.21 mmol/L and a mean HDL cholesterol of 1.23 ± 0.31 mmol/L. Additionally, the study population exhibited a mean SBP of 137.16 ± 15.41 mmHg and a mean DBP of 82.99 ± 10.69 mmHg. Comorbidities were also prevalent, with 18.5% (*n* = 2063) of the participants having diabetes mellitus and 26.3% (*n* = 2939) receiving treatment for hypertension. Notably, 11.2% (*n* = 1248) were on dyslipidemia treatment with statins. The cohort included current smokers (*n* = 2305, 20.6%) and ex-smokers (*n* = 686, 6.1%), underlining the complexity of cardiovascular risk in this population. This comprehensive baseline characterization provided the groundwork for the comparative analysis of eight cardiovascular risk prediction models (Table 1).

### 3.2. Risk Category Distribution

In the comparative analysis of the predicted cardiovascular risk categories, the SCORE2 model categorized a substantial majority of the participants as high-risk (*n* = 7530, 67.4%), whereas the PCE largely identified individuals as low-risk (*n* = 6931, 61%) (Figure 2). The ASSIGN risk score presented a more balanced risk distribution across all three categories. Notably, the RRS and the FRS-hCHD identified a significantly lower number of individuals as high-risk (*n* = 170 and *n* = 833, respectively) compared to models like SCORE2 and the MESA risk score (*n* = 3447). 

### 3.3. Pairwise Agreement Analysis

The heatmap that visualizes the Cohen’s Kappa statistics calculated between each pair of risk prediction models is illustrated in Figure 3. Values range from −1 to 1, with higher values indicating a stronger agreement.

The PCE and AusCVDRisk risk score exhibit a notably high degree of agreement (κ = 0.64), which suggests that these models often categorize patients similarly and could be considered somewhat interchangeable for risk stratification purposes. The MESA and ASSIGN risk scores (κ = 0.4), QRISK3 and AusCVDRisk (κ = 0.51), as well as the QRISK3 score and the PCE (κ = 0.56) demonstrate moderate agreement, implying that they share some similarities in risk categorization but are not entirely interchangeable. Finally, the SCORE2 model exhibits either negative or low agreement with other models, signifying a divergent approach to risk categorization. 

### 3.4. Cluster Analysis: Hierarchical Clustering

The hierarchical clustering dendrogram identified three major clusters of models with similar risk stratification patterns (Figure 4). The first group consists of the MESA and ASSIGN models, indicating a significant similarity in their approach to stratifying cardiovascular risk. The second group, which is larger, includes the AusCVDRisk, PCE, QRISK3, RRS, and FRS-hCHD models. These models, while diverse, share enough commonalities in their risk assessment methodologies to be clustered together, suggesting a broadly aligned perspective in cardiovascular risk prediction. Notably, the SCORE2 model forms a separate and distinct group, indicating a unique approach in its risk stratification criteria compared to the other models. 

### 3.5. Principal Component Analysis (PCA)

PCA was employed to reduce the dimensionality of the risk categories assigned by the eight models for each patient. The aim was to visualize any inherent grouping or separation between the models in a 2D plane. The scatter plot (Figure 5) represents the first two principal components derived from the PCA of the eight cardiovascular risk prediction models. Each point corresponds to a patient, and the coordinates are determined by the risk categories assigned by the models. The scatter plot exhibits a diffuse pattern without distinct clusters. This suggests that the risk prediction models often diverge in their categorization of patients, reinforcing the heterogeneity observed in the pairwise Kappa statistics.

### 3.6. Collective Model Agreement Analysis

Next, we quantified how often all models agree or disagree with classifying the same patient into a specific risk category, which provided an aggregate view of model concordance. The frequency and percentage of cases where a given number of models agree on the risk category for the same patient is presented in Table 2. This analysis revealed that in only 167 cases (1.49%), all eight models agree on the risk category for the same patient. The most common scenarios involve 4 to 7 models agreeing on the risk category, covering roughly 90% of the patients (range: 19.30–23.90%).

## 4. Discussion

The present study elucidates the comparative efficacy of eight widely used cardiovascular risk prediction models in a Lithuanian high cardiovascular risk cohort. Our findings unveil considerable heterogeneity in risk stratification among these models, underscoring the exigency for circumspect application in clinical practice. 

The distribution of risk categories revealed that the SCORE2 model categorized a substantial majority (67.4%) of the participants as high-risk, while the PCE largely (61%) identified individuals as low-risk. The ASSIGN risk score presented a more balanced risk distribution across all three categories. The RRS and FRS-hCHD identified a significantly lower number of individuals as high-risk compared to models like SCORE2 and the MESA risk score. 

The study found that the PCE and AusCVDRisk risk score exhibited a notably high degree of agreement (κ = 0.64), suggesting that these models often categorize patients similarly and could be considered somewhat interchangeable for risk stratification purposes. The MESA and ASSIGN risk scores, as well as the QRISK3 score and the PCE, demonstrated moderate agreement, implying that they share some similarities in risk categorization. On the other hand, the SCORE2 model exhibited either negative or low agreement with other models, signifying a divergent approach to risk categorization. Such discrepancies underscore the varying propensities of these models to categorize individuals into different risk strata, emphasizing the necessity for a cautious interpretation of their predictive capabilities in the LitHiR cohort.

A multitude of factors could have contributed to the observed lack of concordance among the CVD RPMs in our study. Notably, the disparate risk classification thresholds inherent to each tool stand out as a significant factor. Secondly, the discrepancy in model endpoints stems from a varied focus on either specific or broad cardiovascular events, different prediction timeframes (5 or 10 years), and the distinct risk factors incorporated, leading to tailored risk estimations for diverse population subgroups and clinical scenarios across the models [6]. However, since the agreement is based on the ranks rather than the absolute values of the risks, this argument might not be applicable. A poor agreement between CVD risk assessment tools can also occur because of the source cohorts from which the individual risk assessment tools are generated. The SCORE2 tool is derived from multiple European cohorts [9], whereas the ASSIGN model utilizes Scottish data incorporating social deprivation factors [13], the AusCVDRisk is based on Australian populations, the FRS-hCHD emanates from the Framingham Heart Study cohort in the United States [8], the MESA encompasses diverse ethnic groups within the United States [15], the PCE integrates multiple cohorts to derive race- and sex-specific estimations [10], the QRISK3 calculator is developed from a large UK-based cohort integrating a wide range of risk factors [11], and the US-based RRS was formulated through incorporating biochemical variables and a family history of premature coronary heart disease alongside traditional risk factors [12].

Building upon the findings of the comparative analysis, it is crucial to contextualize these results within the broader landscape of cardiovascular risk prediction studies. A systematic review encompassing a comprehensive evaluation of the Framingham risk score, ASSIGN, SCORE, PROCAM, QRISK1 and QRISK2 algorithms, and RRS underscores the variability in performance metrics such as discrimination, calibration, and reclassification across different models and brings attention to the susceptibility of these models to outcome selection and optimism biases, particularly in relation to newer models [16].

Contrary to our SCORE2 findings, the first iteration of the SCORE model was regarded to underestimate the risk of CVD in various populations worldwide [17,18]. By contrast, the Framingham risk score was shown to overestimate the risk of fatal and non-fatal CVD when used in non-US populations [17]. Likewise, the PCE has been reported to overestimate the risk of CVD in Chinese and underestimate the risk in South Asian American populations [19,20]. In addition, the PCE was shown to underpredict observed events in samples with a lower socioeconomic status or with chronic inflammatory diseases, such as HIV, rheumatoid arthritis, or sarcoidosis [21,22,23]. Another large study by Mortensen et al. [24] compared the eligibility criteria for lipid-lowering therapy in the context of primary CVD prevention, utilizing three distinct RPMs: SCORE2, PCE, and QRISK3. Their findings revealed substantial disparities in the proportion of individuals qualifying for class I recommendations for lipid-lowering treatment; only 4% were eligible under the SCORE2 criteria, in contrast to 34% for PCE and 20% for QRISK3. Consequently, it is evident that studies conducted across diverse populations have generally demonstrated a low level of agreement between different risk prediction models. Despite variations in statistical methodologies and demographic variables, the recurrent demonstration of poor concordance between multiple risk models suggests inherent limitations or biases within the tools themselves, warranting further scrutiny and potential refinement. While methodological differences preclude direct comparison, the overarching consensus on the inadequacy of agreement between risk assessment tools cannot be easily dismissed since the observed discrepancies among models could lead to varying clinical decisions, from medication prescriptions to lifestyle recommendations. Therefore, understanding the nuances of each model is crucial for individualized patient care.

### Study Strengths and Limitations

One of the major strengths of this study is its comprehensive comparison of eight different cardiovascular risk prediction models, a scale of analysis seldom achieved in prior investigations. This robust evaluation is further bolstered by a large sample size of 11,174 participants, lending a high level of statistical power and enhancing the reliability and generalizability of the findings. The cohort is particularly diverse in terms of cardiovascular risk profiles, including hypertension, diabetes, and dyslipidemia, which provides a real-world representation of the patients often encountered in clinical practice. Additionally, the methodological rigor is elevated by the application of advanced statistical analyses, including Cohen’s Kappa coefficients for assessing inter-model agreement. Compared to prior studies, we included the newest models, such as SCORE2, QRISK3, the updated AusCVDRisk score, as well as the newest PCE version, which has not yet been demonstrated, further augmenting the contemporary relevance and comprehensive nature of our analysis.

While the use of data from a single large tertiary hospital is often cited as a limitation due to its potential to restrict the external validity of the findings, it can also be considered a strength. This is because a single-center cohort ensures uniformity in data collection methods and eliminates inter-facility variations that could introduce confounding variables. However, this single-center nature remains the most significant limitation of the study, as it may not be fully representative of broader population dynamics. To address this, we have further analyzed the implications of selecting a cohort that solely comprises individuals with a metabolic syndrome. This selection bias could limit the generalizability of our findings, emphasizing the need for future studies to include a more diverse population sample. Specifically, the cohort’s composition of individuals all diagnosed with a metabolic syndrome further narrows the study’s applicability, as these participants may exhibit a distinct cardiovascular risk profile compared to the general population. This specificity underscores the need for the cautious extrapolation of our findings to broader, more heterogeneous populations. The study’s cross-sectional design also precludes the establishment of causality and limits the ability to observe changes in risk predictions over time. Furthermore, the absence of actual cardiovascular event data hampers the validation of the predictive accuracy of the models. The lack of follow-up data analysis is a notable limitation; however, this aspect will be addressed in our future work to assess the predictive characteristics of the aforementioned risk prediction models. Although the large sample size and diversity of risk profiles strengthen the study, the potential for unmeasured confounding factors cannot be entirely dismissed. Lastly, the varying tendencies of the different models to categorize individuals into specific risk categories necessitate a cautious approach in interpreting and applying these models interchangeably.

## 5. Conclusions

The choice of cardiovascular RPM can substantially influence clinical decision-making and patient management. The PCE and AusCVDRisk models exhibited the highest degree of agreement while the SCORE2 model consistently exhibited low or negative agreement with other models, signifying a divergent approach to risk categorization with only 1.49% of cases where all eight models agree on the risk category for the same patient. These discrepancies underscore the importance of adopting a more refined approach to using RPMs, necessitating further research to evaluate their predictive accuracy in the Lithuanian population.

## Figures and Tables

**Figure 1 jcm-13-01806-f001:**
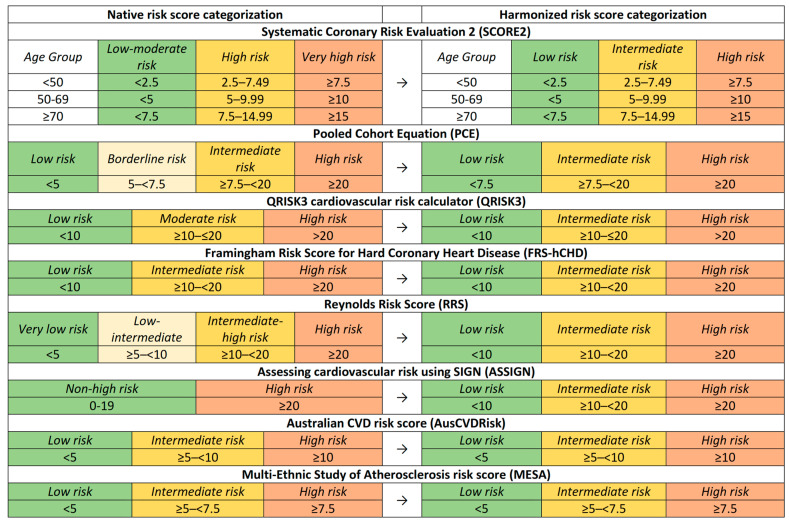
Comparative display of native and harmonized cardiovascular risk categorizations post-adjustment.

**Figure 2 jcm-13-01806-f002:**
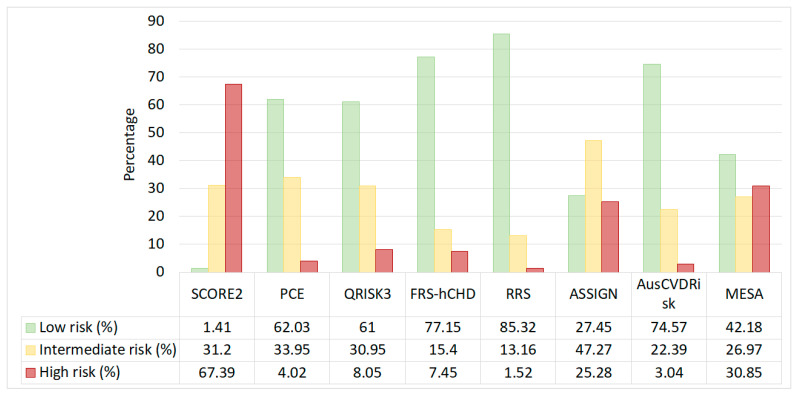
Distribution of cardiovascular risk categories across eight cardiovascular risk prediction models.

**Figure 3 jcm-13-01806-f003:**
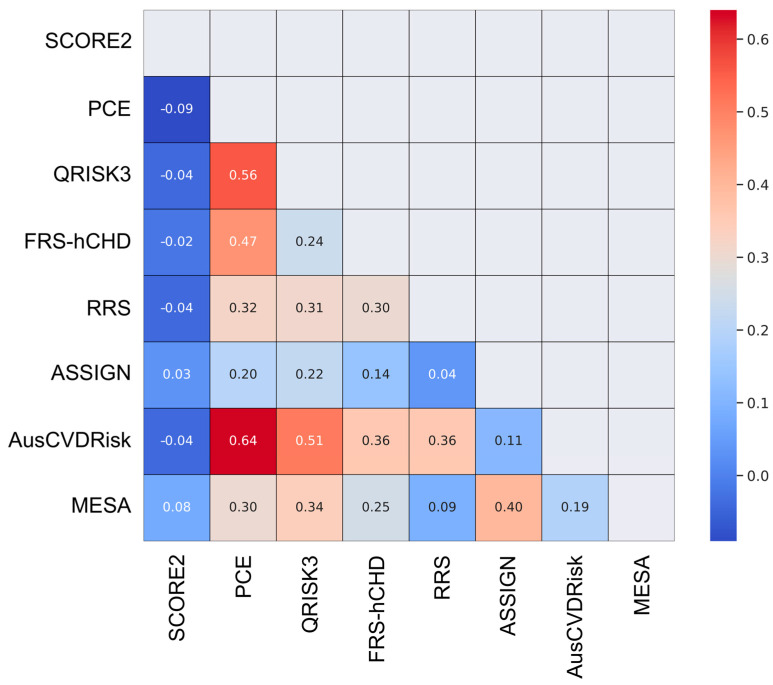
Heatmap demonstrating pairwise agreement among eight cardiovascular risk prediction models.

**Figure 4 jcm-13-01806-f004:**
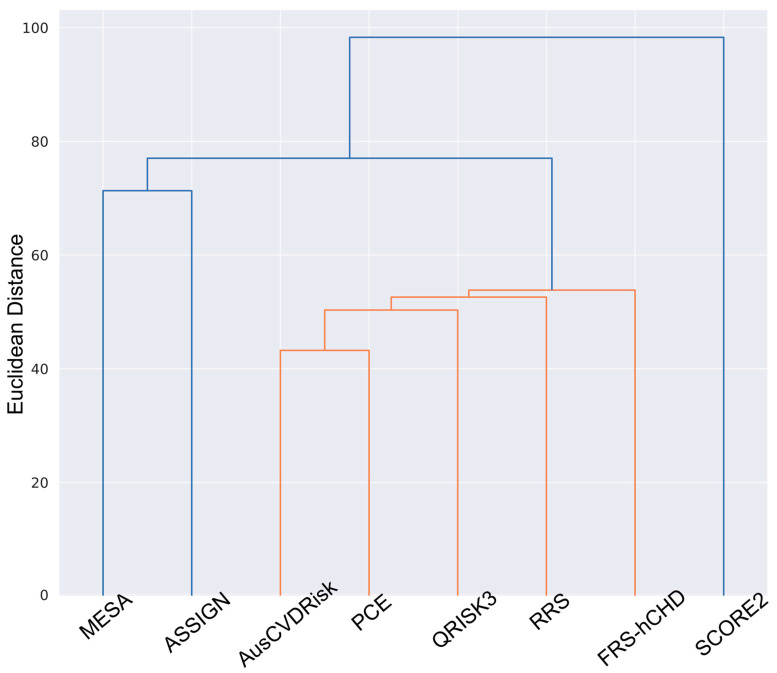
Dendrogram of hierarchical clustering to assess similarities among cardiovascular risk prediction models.

**Figure 5 jcm-13-01806-f005:**
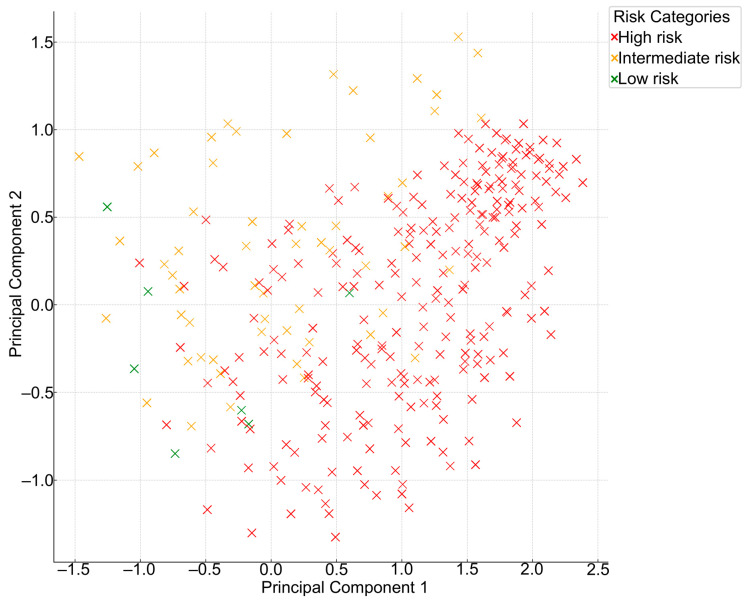
Principal component analysis of patient risk categories across eight cardiovascular models.

**Table 1 jcm-13-01806-t001:** Baseline characteristics of the study population (*n* = 11,174).

Characteristics	
Gender—*n* (%)	Female 6527 (58.4)
Age, years—mean (SD)	53.49 (6.47)
Body mass index, kg/m^2^—mean (SD)	31.57 (4.46)
Total cholesterol, mmol/L—mean (SD)	6.17 (1.37)
LDL cholesterol, mmol/L—mean (SD)	3.98 (1.21)
HDL cholesterol, mmol/L—mean (SD)	1.23 (0.31)
Triglycerides, mmol/L—mean (SD)	2.11 (1.5)
Fasting glucose, mmol/L—mean (SD)	6.31 (1.49)
Creatinine, µmol/L—mean (SD)	71.69 (12.79)
Systolic blood pressure, mmHg—mean (SD)	137.16 (15.41)
Diastolic blood pressure, mmHg—mean (SD)	82.99 (10.69)
Diabetes mellitus—*n* (%)	2063 (18.5)
Hypertension treatment—*n* (%)	2939 (26.3)
Dyslipidemia treatment (statins)—*n* (%)	1248 (11.2)
Antiplatelet treatment—*n* (%)	30 (0.3)
Current smoker—*n* (%)	2305 (20.6)
Ex-smoker—*n* (%)	686 (6.1)

HDL—high-density lipoprotein; LDL—low-density lipoprotein; SD—standard deviation.

**Table 2 jcm-13-01806-t002:** Frequency and percentage of agreement among models for patient risk categories.

Number of Models Agreeing	Number of Patients	Percentage of Patients (%)
3	974	8.72
4	2671	23.90
5	2157	19.30
6	2594	23.21
7	2611	23.37
8	167	1.49

## Data Availability

The data underlying this article cannot be shared publicly due to the stipulations of the bioethical approval which precludes unrestricted data dissemination and confines access to individuals delineated within the approval framework. The data will be shared upon reasonable request to the corresponding author.

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
