# Peer review of "Navigating the Landscape of Cardiovascular Risk Scores: A Comparative Analysis of Eight Risk Prediction Models in a High-Risk Cohort in Lithuania"

_jcm, 2024, doi:10.3390/jcm13061806_

Round 1

Reviewer 1 Report

Comments and Suggestions for Authors

This study is relevant in that it attempts to compare eight risk prediction models in a specifically chosen high risk cohort in a high risk country, Lithuania. SCORE2 produced the highest risk estimates while PCE categorised most as at low risk. It should be noted that the derivation cohorts showed considerable heterogeneity, making close agreement between the scores improbable.

Of note is that the models were tested on a specific cohort  of 11,174 individuals from a single hospital, all with metabolic syndrome. This is a problem- the risk models studied generally do not include the metabolic syndrome, and the applicability of the findings to the general Lithuanian population must be open to question. 

Table 1 indicates that the study population were indeed at high risk- BMI 32, cholesterol 6.2, , LDL cholesterol 3.8, triglycerides 2.11, glucose 6.3, BP 137/83, diabetes 18.5%. Given this, the assignment by SCORE2 of 67% to a high risk category is hardly surprising, even if higher than all others which gave a high risk score of between ann extremely improbable 1.52% and 32%.

While the methods of assessment are of interest, it may be that the data have been substantially over-analysed given the heterogeneity of the studies. The discussion is not adequate and needs tp stress thew limitations more, particularly with regards to the highly selected test cohort and hence doubts about the generalisability to the general population and the considerable heterogeneity of the studies assessed. Given the authors knowledge of the situation in Lithuania, and the study of a high risk cohort within a high risk country, , which risk estimation system do they feel likely to be appropriate? Of note, the only system to be re-calibrated to 4 risk regions in Europe is SCORE2. Please explain why other risk estimation systems may be more appropriate.

Comments on the Quality of English Language

Adequate

Author Response

Dear Reviewer,

Thank you for your comprehensive review and insightful comments on our manuscript. Your feedback has been invaluable in guiding us towards refining our study for greater clarity, relevance, and impact. Below, we address each of your points and outline the changes we have implemented in response.

Comment 1. Regarding Discussion of Limitations: “Of note is that the models were tested on a specific cohort  of 11,174 individuals from a single hospital, all with metabolic syndrome. This is a problem- the risk models studied generally do not include the metabolic syndrome, and the applicability of the findings to the general Lithuanian population must be open to question.”

We agree with the highlighted issue and as a result, we believe that the discussion on the study's limitations needs to be strengthened. The revised discussion section now comprehensively addresses these aspects, particularly focusing on the selection bias inherent in our chosen cohort and its implications for generalizability.

The rewritten section:

While the use of data from a single large tertiary hospital is often cited as a limitation due to its potential to restrict the external validity of the findings, it can also be considered a strength. This is because a single-centre cohort ensures uniformity in data collection methods and eliminates inter-facility variations that could introduce confounding varia-bles. However, this single-centre nature remains the most significant limitation of the study, as it may not be fully representative of broader population dynamics. To address this, we have further analyzed the implications of selecting a cohort solely comprised of individuals with metabolic syndrome. This selection bias could limit the generalizability of our findings, emphasizing the need for future studies to include a more diverse popula-tion sample. Specifically, the cohort's composition of individuals all diagnosed with met-abolic syndrome further narrows the study's applicability, as these participants may ex-hibit a distinct cardiovascular risk profile compared to the general population. This speci-ficity underscores the need for cautious extrapolation of our findings to broader, more het-erogeneous populations. The study's cross-sectional design also precludes the establish-ment of causality and limits the ability to observe changes in risk predictions over time. Furthermore, the absence of actual cardiovascular event data hampers the validation of the predictive accuracy of the models. The lack of follow-up data analysis is a notable lim-itation; however, this aspect will be addressed in our future work to assess the predictive characteristics of the aforementioned risk prediction models. Although the large sample size and diversity of risk profiles strengthen the study, the potential for unmeasured con-founding factors cannot be entirely dismissed. Lastly, the varying tendencies of the differ-ent models to categorize individuals into specific risk categories necessitate a cautious approach in interpreting and applying these models interchangeably.

Comment 2. Regarding categorization data: “Table 1 indicates that the study population were indeed at high risk- BMI 32, cholesterol 6.2, , LDL cholesterol 3.8, triglycerides 2.11, glucose 6.3, BP 137/83, diabetes 18.5%. Given this, the assignment by SCORE2 of 67% to a high risk category is hardly surprising, even if higher than all others which gave a high risk score of between ann extremely improbable 1.52% and 32%.”

We appreciate your observation that the SCORE2 model classified a significant proportion of our high-risk cohort as being at high/very high risk, aligning with the clinical characteristics presented in Table 1. The striking contrast in risk categorization between SCORE2 and other models, where the same cohort was predominantly classified as low risk, underscores a critical inquiry into the efficacy and appropriateness of these risk prediction models (RPMs) within specific populations. Acknowledging this discrepancy, we aim to delve deeper into the predictive accuracy of these RPMs in our subsequent analysis, leveraging outcome data to provide a more definitive evaluation. This forthcoming research could offer valuable insights into the relative performance of each model, potentially guiding clinicians towards the most effective RPM for their patient populations.

Comment 3. Appropriateness of Risk Estimation Systems for Lithuania: “While the methods of assessment are of interest, it may be that the data have been substantially over-analysed given the heterogeneity of the studies. The discussion is not adequate and needs tp stress thew limitations more, particularly with regards to the highly selected test cohort and hence doubts about the generalisability to the general population and the considerable heterogeneity of the studies assessed. Given the authors knowledge of the situation in Lithuania, and the study of a high risk cohort within a high risk country, , which risk estimation system do they feel likely to be appropriate? Of note, the only system to be re-calibrated to 4 risk regions in Europe is SCORE2. Please explain why other risk estimation systems may be more appropriate.”

Thank you for your insightful query regarding the appropriateness of different risk estimation systems for our study cohort in Lithuania. Given the limitations of our analysis, which did not include outcome data, pinpointing the most suitable risk prediction model is challenging. Our findings indicate a considerable variation in risk stratification across the models examined, with SCORE2 often categorizing individuals into higher risk categories. This propensity could potentially lead to an overestimation of risk, which is particularly concerning from a clinical perspective.

In contexts where reclassification into different risk groups directly impacts treatment decisions, such overestimation could detract from the personalized medicine approach, homogenizing treatment pathways rather than tailoring them to individual patient profiles. Clinicians, aware of SCORE2's tendency to assign a high-risk status, may bypass its use, anticipating its outcome. This practice underscores the necessity for RPMs that offer more nuanced risk differentiation, enabling truly personalized treatment plans.

While SCORE2's recalibration for European regions, including Lithuania, is noteworthy, our analysis suggests the need for further refinement of RPMs to better serve diverse patient populations, aligning with the goal of personalized healthcare.

In addition, we wish to inform you that Figures 1 and 2 have been meticulously corrected and the abstract has been rewritten, adopting a structure and content that more closely adhere to the journal's guidelines.

We are grateful for the opportunity to improve our manuscript based on your feedback and believe these revisions have enhanced the quality and relevance of our work. We look forward to your thoughts on the revised manuscript.

Kind regards,

Petras Navickas

Reviewer 2 Report

Comments and Suggestions for Authors

In the current study, the authors aimed to compare the performance of several CVD predictions models in Lithuania. This was done using cross-sectional data from a single, tertiary university hospital. Follow-up data was not available to assess the external validation of the models. The study is generally well written and offers some insight in the potential effect of applying any of the risk models in Lithuania. However, I do have several concerns:

-        Lack of follow-up data limits the study to only assess the predicted risks, whereas no guidance can be given on which risks models would be most accurate in their use, or most useful the Lithuanian clinicians. The main conclusion is that the SCORE2 assigns risk categories quite differently from the other models, which could have been expected given that this is the only model that reflects the (a lot) higher incidence of cardiovascular disease in Eastern Europe and is the only model that has age-specific risk cut-offs, which also leads to very different patterns of individuals classified as high risk.

-        The conclusions is that a more refined approach is needed towards prediction models in Lithuania. I think that is hard to conclude based on this data, as it is unclear whether any of the current models work well. I do agree completely with their conclusion that adequate validation in this part of Europe of either of these include models is very needed, as these validations have been scarce so far.

-        The population consists of individuals with metabolic syndrome, which is only a small fraction of the target population of these models. This limits the generalizability to the population at large.

-        For the current study, the comparison is mostly based on categorization into risk categories. However, as can be seen from Figure 1, the cut-offs from each model differ a lot. It would be interesting to also see a comparison between the continuous risk predictions (given the same endpoints, which is of course not true for all models)

Author Response

Dear Reviewer,

Thank you for your comprehensive review and insightful comments on our manuscript. Your feedback has been invaluable in guiding us towards refining our study for greater clarity, relevance, and impact. Below, we address each of your points and outline the changes we have implemented in response.

Comment 1. Lack of follow-up data: “Lack of follow-up data limits the study to only assess the predicted risks, whereas no guidance can be given on which risks models would be most accurate in their use, or most useful the Lithuanian clinicians. The main conclusion is that the SCORE2 assigns risk categories quite differently from the other models, which could have been expected given that this is the only model that reflects the (a lot) higher incidence of cardiovascular disease in Eastern Europe and is the only model that has age-specific risk cut-offs, which also leads to very different patterns of individuals classified as high risk.”

Thank you for your valuable feedback regarding the limitations of our study due to the absence of follow-up data. We acknowledge that this restricts our analysis to assessing predicted risks without evaluating the models' accuracy or utility for Lithuanian clinicians.

We would like to clarify that we possess follow-up data for the study's population. Building on the findings of this paper, our subsequent work will focus on analyzing the predictive characteristics of the risk prediction models discussed. Given the fundamentally different methodology required for such an analysis, we believe it merits a separate publication.

Although it might turn out to be the case that the SCORE2 model might exhibit superior predictive accuracy, its utility from a clinical perspective warrants careful consideration. Specifically, our concern lies in the potential over-reliance on categorical risk groups provided by SCORE2, which could lead to a homogenized approach to patient care, moving away from the personalized medicine paradigm. This is particularly problematic if the model categorizes a disproportionately high number of individuals as high/very high risk for CVD, diminishing the opportunity for nuanced patient-specific intervention strategies.

We agree that the discussion on the study's limitations needed to be strengthened. Therefore, we have revised the discussion section, which now comprehensively addresses these aspects.

The rewritten section:

While the use of data from a single large tertiary hospital is often cited as a limitation due to its potential to restrict the external validity of the findings, it can also be considered a strength. This is because a single-centre cohort ensures uniformity in data collection methods and eliminates inter-facility variations that could introduce confounding variables. However, this single-centre nature remains the most significant limitation of the study, as it may not be fully representative of broader population dynamics. To address this, we have further analyzed the implications of selecting a cohort solely comprised of individuals with metabolic syndrome. This selection bias could limit the generalizability of our findings, emphasizing the need for future studies to include a more diverse population sample. Specifically, the cohort's composition of individuals all diagnosed with metabolic syndrome further narrows the study's applicability, as these participants may exhibit a distinct cardiovascular risk profile compared to the general population. This specificity underscores the need for cautious extrapolation of our findings to broader, more heterogeneous populations. The study's cross-sectional design also precludes the establishment of causality and limits the ability to observe changes in risk predictions over time. Furthermore, the absence of actual cardiovascular event data hampers the validation of the predictive accuracy of the models. The lack of follow-up data analysis is a notable limitation; however, this aspect will be addressed in our future work to assess the predictive characteristics of the aforementioned risk prediction models. Although the large sample size and diversity of risk profiles strengthen the study, the potential for unmeasured confounding factors cannot be entirely dismissed. Lastly, the varying tendencies of the different models to categorize individuals into specific risk categories necessitate a cautious approach in interpreting and applying these models interchangeably.

Comment 2. Regarding conclusions: “The conclusions is that a more refined approach is needed towards prediction models in Lithuania. I think that is hard to conclude based on this data, as it is unclear whether any of the current models work well. I do agree completely with their conclusion that adequate validation in this part of Europe of either of these include models is very needed, as these validations have been scarce so far.”

We acknowledge your perspective regarding the preliminary nature of our conclusions on the need for a more refined approach towards prediction models in Lithuania. While we concur that the absence of outcome data in our current analysis limits our ability to definitively endorse the most appropriate model, we emphasize that our findings—particularly the skewed risk category distribution observed with SCORE2—highlight a critical issue. Even if subsequent analyses were to establish SCORE2 as the most accurate risk prediction model (RPM), this would not mitigate the concern that its risk category intervals may disproportionately classify a majority of patients into the highest risk categories. This observation underscores the necessity for a tailored approach in evaluating and possibly adjusting RPMs to better reflect the cardiovascular risk landscape in Lithuania, ensuring both accuracy and practical utility in clinical decision-making.

Comment 3. Regarding study’s population: “The population consists of individuals with metabolic syndrome, which is only a small fraction of the target population of these models. This limits the generalizability to the population at large.”

We concur with your observation regarding the study population's composition primarily of individuals with metabolic syndrome. We acknowledge that this cohort represents only a subset of the broader target population for these risk models, potentially limiting the generalizability of our findings. To address this concern, we have enhanced our discussion section by elaborating on the limitations regarding the study's generalizability.

Comment 4. Regarding cut-off values: “For the current study, the comparison is mostly based on categorization into risk categories. However, as can be seen from Figure 1, the cut-offs from each model differ a lot. It would be interesting to also see a comparison between the continuous risk predictions (given the same endpoints, which is of course not true for all models)”

We appreciate your suggestion regarding the comparison between continuous risk predictions across different models, recognizing the significant variation in cut-offs highlighted in Figure 1. Indeed, such an analysis, comparing continuous risk scores while accounting for the diverse endpoints each model targets, presents a valuable opportunity to deepen our understanding of these tools' predictive capabilities. Acknowledging this, we plan to extend our research to include an evaluation of the predictive accuracy of the RPMs mentioned, incorporating continuous risk predictions into our future analysis. This forthcoming work aims to offer a more nuanced comparison and potentially bridge the gaps identified in the current study's categorical risk assessment approach.

In addition, we wish to inform you that Figures 1 and 2 have been meticulously corrected and the abstract has been rewritten, adopting a structure and content that more closely adhere to the journal's guidelines.

We are grateful for the opportunity to improve our manuscript based on your feedback and believe these revisions have significantly enhanced the quality and relevance of our work. We look forward to your thoughts on the revised manuscript.

Kind regards,

Petras Navickas

Round 2

Reviewer 1 Report

Comments and Suggestions for Authors

Now acceptable